DATA RELEASE

# Occurrence records and metadata for sand flies (Diptera, Psychodidae, Phlebotominae) collected in the lands of indigenous people in the Brazilian Amazon

Paloma Helena Fernandes Shimabukuro[1,2,*], Daniel Rocha Cangussu Alves[3], Jéssica Adalia Costa Barros[2], Luiz Otavio Cordeiro Nascimento[4], Luke Anthony Baton[5], Maíra Posteraro Freire[4,6], Manoel Edson Medeiros da Silva[3], Mauro Diego Gobira Guimarães de Assis[2], Sofia Ferreira Morais[2], Tiago Silva da Costa[7], Veracilda Ribeiro Alves[8] and Eduardo Stramandinoli Moreno[4,9]

1 Coleção de Flebotomíneos (FIOCRUZ/COLFLEB), Fiocruz, Belo Horizonte, Brazil
2 Grupo de Estudos em Leishmanioses, Fiocruz, Belo Horizonte, Brazil
3 Frente de Proteção Etnoambiental Médio Purus/Coordenação-Geral de Índios Isolados e de Recente Contato – CGIIRC/Fundação Nacional do Índio. Lábrea, Amazonas, Brazil
4 Distrito Sanitário Especial Indígena Amapá e Norte do Pará – Secretaria Especial de Saúde Indígena - Ministério da Saúde, Macapá, Amapá, Brazil
5 50 Rowntree Way, Saffron Walden, Essex CB11 4DL, UK
6 Programa de Pós-Graduação em Saúde, Sociedade e Endemias na Amazônia, Instituto Leônidas e Maria Deane, Fundação Oswaldo Cruz/Universidade Federal do Amazonas – Manaus, Amazonas, Brazil
7 Laboratório de Arthropoda (Arthrolab), Universidade Federal do Amapá (Unifap), Macapá, Amapá, Brazil
8 Instituto Evandro Chagas, Ananindeua, Pará, Brazil
9 Programa de Pós-Graduação: Sociedade, Natureza e Desenvolvimento – Universidade Federal do Oeste do Pará, Santarém, Pará, Brazil

**Submitted:** 28 February 2022

\* Corresponding author. E-mail: phfs@yahoo.com

Preprint submitted at https://doi.org/10.1590/SciELOPreprints.3879

Included in the series: *Vectors of human disease* (https://doi.org/10.46471/GIGABYTE_SERIES_0002)

## ABSTRACT

To contribute to knowledge of the epidemiology of American cutaneous leishmaniasis (ACL) among indigenous people living in sylvatic regions, we studied the sand fly fauna collected in areas of disease transmission in the Brazilian Amazon. Here we report two datasets comprising occurrence data for sand flies from the Suruwaha Indigenous Land in the state of Amazonas collected in 2012–2013, and the Wajãpi Indigenous Land in the state of Amapá collected in 2013–2014. Sand flies were collected using unbaited CDC-like light traps at various sites within each study area and were identified to either genus or species-level by taxonomists with expertise in Amazonian fauna. A total of 4,646 records are reported: 1,428 from the Suruwaha and 3,218 from the Wajãpi. These records will contribute to a better understanding of ACL transmission dynamics, as well as the distribution of insect vectors, in these areas.

**Subjects** Animal and Plant Sciences, Biodiversity, Taxonomy

## DATA DESCRIPTION

Leishmaniases are diseases caused by various species of the protozoan parasite genus *Leishmania*, which are transmitted between humans, and wild and domestic vertebrate animals, by the bites of blood-feeding female sand flies [1]. In Brazil, American cutaneous leishmaniasis (ACL) is an endemic disease, but little is known about its impact on indigenous human populations, especially those people who live in remote areas and have little contact with non-indigenous people living outside their territorial lands. To investigate the sand fly fauna present during outbreaks of ACL among the Suruwaha and the Wajãpi indigenous people that occurred between 2012 and 2014, we carried out fieldwork to collect these insects and identify potential vectors. Insects were identified by experienced taxonomists using keys available in the literature [2, 3].

Our datasets comprise sand fly occurrence data from: (i) the Suruwaha Indigenous Land (SIL) in the south of the state of Amazonas collected between 2012 and 2013; and (ii) the Wajãpi Indigenous Land (WIL) in the state of Amapá collected between 2013 and 2014. The datasets reported here are the metadata for each individual sand fly specimen collected during the fieldwork and include 41 Darwin Core Standard (DwC) terms [4] for the Suruwaha dataset and 39 for the Wajãpi dataset. All mandatory fields are present and have undergone screening in the integrated publishing toolkit (IPT) of the Fundação Oswaldo Cruz (FIOCRUZ). Metadata fields are also available on the online pages [5, 6].

For each individual sand fly specimen, our dataset includes fields describing their: (i) taxonomy (kingdom, phylum, class, order, family, genus, specificEpithet, verbatimIdentification, infraspecificEpithet, scientificName, scientificNameAuthorship, taxonRank); (ii) collection details, including the collectors (recordedBy: Shimabukuro PHF; Stumpp RGAV; Medeiros MES.; Alves DRC; Moreno ES; Freire M P; Nascimento LOC), collection date, trapping method, trap identification number, and collection site description (verbatimEventDate, eventTime, habitat, samplingProtocol); (iii) geolocation data (stateProvince, county, locality, locationRemarks, verbatimLatitude, verbatimLongitude, decimalLatitude, decimalLongitude, geodeticDatum); and (iv) catalog reference data (otherCatalogNumbers).

Our datasets [5, 6] are publically available in the Sistema de Informação sobre a Biodiversidade Brasileira (SiBBr). This is an online platform integrating data and information about biodiversity and ecosystems and is the Brazilian Node of the Global Biodiversity Information Facility (GBIF), an internationally-recognized resource for collation of biological occurrence data [7].

### Context

ACL is a serious public health problem in Brazil, with 203,406 cases recorded between 2010 and 2020. The administrative North Region of the country, which includes the Amazon biome, currently accounts for 42% of all cases recorded within Brazil [8].

Health and disease conditions among indigenous people in Brazil are poorly understood. Government databases have big gaps in the information they hold on indigenous morbidity, mortality, disease notification and so on, which prevent construction of the most basic sociodemographic and health indicators. However, the few studies on the public health of indigenous populations in Brazil reveal marginalization and vulnerability, which translate into low quality of life, difficulty in accessing services, and low health indicators [9].

The different transmission patterns of ACL also make it a difficult disease to control and understand. This is especially true in the Brazilian Amazon, which has the greatest diversity of *Leishmania*-species parasites, their insect vectors and vertebrate hosts, in the neotropical region [1].

Our two datasets comprised sand fly occurrence data from fieldwork undertaken by the authors during the ACL outbreaks in the SIL and WIL.

## Suruwaha dataset

Transmission of ACL south of the Amazon River system is poorly understood [10, 11]. Although it has often been stated that human *Leishmania* infection is either rare or absent south of the Negro and Amazon Rivers [10, 12], the incidence of leishmaniasis in humans in some of these areas is equivalent to the incidence north of the Negro and Amazon Rivers [11, 13]. Guerra *et al.* (2011) [14] described the epidemiology of mucosal leishmaniasis (ML) south of the Amazon River, and not only found a high prevalence of this form of the disease, but also a distribution of *Leishmania* species similar to that found north of the Amazon River. However, the etiologic agent of cutaneous leishmaniasis (CL) has not been identified, and the source of sylvatic infection and the vectors involved in transmission are not known south of the Amazon River system [10, 15].

The Suruwaha is a population of approximately 170 people [16], whose only contact with non-indigenous people is through health workers and members of the Fundação Nacional do Índio (FUNAI), the Brazilian government agency responsible for protecting indigenous people. The Suruwaha live in a remote location, far from urban areas and the manufacture of utensil. Their main activities are agriculture and hunting.

We conducted entomological research within the SIL to contribute to understanding the transmission dynamics of ACL. This research was a response an increase in the number of cases of ACL recorded among the Suruwaha between 2010 and 2012. Our investigation found that this increase was probably related to the distribution of flashlights to the entire community in late 2010, as requested by the Suruwaha themselves. Use of these flashlights led to changes in the Suruwaha's hunting habits, from strictly diurnal to nocturnal, coinciding with the generally crepuscular and nocturnal biting activity of the blood-feeding female sand flies that transmit ACL.

The SIL is remote. Our entomological survey was performed by different teams to take advantage of routine operations by FUNAI. A total of 1,428 sand fly specimens were identified to either genus or species-level. Some specimens could not be identified to species level because they are morphologically indistinguishable, morphological structures were not clearly visible in the slide, or the specimens lost appendages important for morphological identification. Ten genera and 33 species were collected (Table 1), of which the genera *Trichophoromyia* was predominant (57%), followed by *Psychodopygus* (20%) and *Nyssomyia* (5%). Among the most abundant species were *T. ubiquitalis* (25%), *T. octavioi* (9%) and *Ps. davisi* (8%). A total of 667 female and 760 male specimens were collected. One specimen lost head and terminalia so could not be sexed.

## Wajãpi dataset

The Wajãpi are a group of about 1,200 people [17], distributed in 80 small villages across their territory. All families subsist by agriculture, fishing, hunting and gathering, periodically changing the location of their villages to allow ecological recovery of the areas

**Table 1.** Species of sand flies collected in the Suruwaha Indigenous Land, municipality of Tapauá, state of Amazonas, between 2012 and 2013.

| Species | Number of specimens |
|---|---|
| *Trichophoromyia ubiquitalis* | 634 |
| *Trichophoromyia* sp. | 146 |
| *Trichophoromyia octavioi* | 133 |
| *Psychodopygus davisi* | 123 |
| *Psychodopygus chagasi* | 80 |
| *Psychodopygus claustrei* | 42 |
| *Bichromomyia flaviscutellata* | 42 |
| *Trichophoromyia melloi* | 41 |
| *Nyssomyia antunesi* | 33 |
| *Psychodopygus ayrozai* | 23 |
| *Nyssomyia fraihai* | 22 |
| *Psychodopygus* sp. | 13 |
| *Nyssomyia* sp. | 10 |
| *Psychodopygus paraensis* | 10 |
| *Psathyromyia* sp. | 7 |
| *Psychodopygus geniculatus* | 6 |
| *Sciopemyia sordellii* | 6 |
| *Evandromyia saulensis* | 5 |
| *Evandromyia* sp. | 6 |
| *Lutzomyia* sp. | 6 |
| *Sciopemyia* sp. | 4 |
| *Trichophoromyia brachipyga* | 4 |
| *Psathyromyia barrettoi barrettoi* | 4 |
| *Evandromyia bacula* | 3 |
| *Evandromyia begonae* | 3 |
| *Lutzomyia evangelistai* | 3 |
| *Psathyromyia abunaensis* | 2 |
| *Psychodopygus hirsutus* | 2 |
| *Psychodopygus llanosmartinsi* | 2 |
| *Psychodopygus amazonensis* | 1 |
| *Psathyromyia aragaoi* | 1 |
| *Viannamyia furcata* | 1 |
| *Psathyromyia inflata* | 1 |
| *Psychodopygus lainsoni* | 1 |
| *Bichromomyia olmeca nociva* | 1 |
| *Bichromomyia olmeca reducta* | 1 |
| *Psathyromyia runoides* | 1 |
| *Sciopemyia servulolimai* | 1 |
| *Viannamyia tuberculata* | 1 |
| *Trichopygomyia* sp. | 1 |
| *Trichopygomyia wagleyi* | 1 |
| Total | 1,428 |

they have occupied. Access to the villages can be via road, rivers and streams crossing their territory, and open trails through the forest. The resumption of the traditional model of Wajãpi occupation and dispersal for territorial exploitation, which maintains the quality of life of the people and, at the same time, guarantees territorial surveillance, was essential to guarantee the pattern of abundance that the Wajãpi consider adequate [18]. In the last 10 years, the activity of large mining companies, and the building of hydroelectric plants, in the environment surrounding the WIL threatens the quality of life of the Wajãpi through forest degradation, environmental contamination, and the insertion of the community in a context of great social vulnerability, coinciding with the increased incidence of pests in crops and endemic diseases, such as malaria and leishmaniasis.



Our dataset comprises sand fly records resulting from the activities of a project to evaluate the main risk factors for the disease, using an interdisciplinary approach. In 2012, there was an outbreak of leishmaniasis in the WIL affecting more than 20 people in seven different villages (data provided by SESAI). This outbreak triggered a broader investigation that was conducted between 2012 and 2015 to understand the causal processes related to the local epidemiological context, to propose measures to prevent and control the disease among the Wajãpi. The investigation unfolded in a multi-institutional project with the partnership of institutions related to health services, such as: Distrito Sanitário Especial Indígena (DSEI) Amapá e Norte do Pará, Secretaria Municipal de Saúde de Pedra Branca do Amapari, Coordenadoria de Vigilância em Saúde do Amapá, Laboratório Central de Saúde Pública do Amapá, Instituto René Rachou (Fiocruz-MG), Universidade Federal do Amapá, Universidade de São Paulo, and Universidade Federal do Oeste do Pará. Teams with different expertise were formed: entomologists, biologists, veterinarians, laboratory technicians, anthropologists, epidemiologists, nurses, and so on.

From the beginning, researchers were faced with a complex environmental and intercultural context, in which the simple intensification of standard measures recommended by health services proved to be insufficient – or even antagonistic in relation to the health principles and concepts of the indigenous group. Poor access to villages made it difficult to map and record possible transmission sites. The long incubation period of the ACL, the complex mobility profile through their territory of the Wajãpi and the limitations of the health services, for example, lack of trained personnel, lack of proper funding and infrastructure, made the search for transmission foci, according to the surveillance standards recommended by the Ministry of Health – based on the search for individual risk factors, an unpromising task.

The need for a broader and more sensitive assessment from the ecological, cultural and ethnic point of view of this epidemiological context unfolded in a review – not only of the objectives and methods of providing services in indigenous health and of epidemiological research and conduct, but also of an epistemological reconstruction of the process of understanding causality in health-disease and its inseparable relationship with social and ecological processes, their different scales and dimensions of perception. Therefore, data on the occurrence of ACL vectors was paramount.

A total of 3,218 specimens (Table 2) were identified to either genus or species-level. Again, some species could not be identified to the species level since they were morphologically indistinguishable, morphological structures were not clearly visible in the slide, or the specimens lost appendages important for morphological identification. The most abundant genera were *Trichophoromyia* (20%), *Nyssomyia* (13%) and *Psathyromyia* (11%). The most abundant species were *T. brachipyga* (14%), followed by *Pa. dreisbachi* (9%) and *Nyssomyia pajoti* (6%). A total of 1,938 female and 1,276 male specimens were collected. Four specimens lost head and terminalia and could not be sexed.

## METHODS

Specimens were georeferenced with the aid of a Garmin® 62s global positioning system (GPS) device.

**Table 2.** Species of sand flies collected in the Wajãpi Indigenous Land, municipality of Pedra Branca do Amapari, state of Amazonas, between 2013 and 2014.

| Species | Number of specimens |
|---|---|
| *Trichophoromyia brachipyga* | 456 |
| *Trichophoromyia* sp. | 345 |
| *Psathyromyia dreisbachi* | 298 |
| *Nyssomyia pajoti* | 204 |
| *Pressatia* sp. | 198 |
| *Evandromyia infraspinosa* | 195 |
| *Trichophoromyia ubiquitalis* | 173 |
| *Nyssomyia umbratilis* | 169 |
| *Trichopygomyia trichopyga* | 165 |
| *Bichromomyia flaviscutellata* | 128 |
| *Psychodopygus davisi* | 112 |
| *Sciopemyia sordellii* | 110 |
| *Evandromyia brachyphalla* | 78 |
| *Trichopygomyia* sp. | 71 |
| *Evandromyia* sp. | 53 |
| *Psychodopygus* sp. | 47 |
| *Psychodopygus geniculatus* | 33 |
| *Evandromyia monstruosa* | 33 |
| *Nyssomyia antunesi* | 31 |
| *Psathyromyia inflata* | 29 |
| *Nyssomyia* sp. | 26 |
| *Psychodopygus hirsutus* | 22 |
| *Psathyromyia* sp. | 16 |
| *Evandromyia andersoni* | 15 |
| *Psathyromyia aragaoi* | 14 |
| *Viannamyia furcata* | 12 |
| *Psychodopygus paraensis* | 12 |
| *Micropygomyia rorotaensis* | 12 |
| *Psychodopygus claustrei* | 11 |
| *Psychodopygus maripaensis* | 11 |
| *Migonemyia migonei* | 10 |
| *Nyssomyia whitmani* | 9 |
| *Micropygomyia* sp. | 8 |
| *Evandromyia pinottii* | 8 |
| *Viannamyia tuberculata* | 8 |
| *Pintomyia damascenoi* | 7 |
| *Sciopemyia fluviatilis* | 7 |
| *Brumptomyia pentacantha* | 7 |
| *Brumptomyia* sp. | 6 |
| *Psychodopygus corossoniensis* | 6 |
| *Sciopemyia* sp. | 5 |
| *Pressatia choti* | 5 |
| *Psathyromyia dendrophyla* | 5 |
| *Trichopygomyia depaquiti* | 5 |
| *Trichophoromyia ininii* | 5 |
| *Bichromomyia* sp. | 4 |
| *Psathyromyia barrettoi barrettoi* | 3 |
| *Psathyromyia lutziana* | 3 |
| *Psathyromyia pradobarrentosi* | 3 |
| *Lutzomyia sherlocki* | 3 |
| *Lutzomyia* sp. | 2 |
| *Psathyromyia abunaensis* | 2 |
| *Pressatia trispinosa* | 2 |
| *Psathyomyia abonnenci* | 1 |
| *Psychodopygus amazonensis* | 1 |



| Species | Number of specimens |
|---|---|
| *Nyssomyia anduzei* | 1 |
| *Evandromyia* sp. de Baduel | 1 |
| *Evandromyia begonae* | 1 |
| *Psathyromyia bigeniculada* | 1 |
| *Brumptomyia brumpti* | 1 |
| *Migonemyia bursiformis* | 1 |
| *Lutzomyia gomezi* | 1 |
| *Sciopemyia nematoducta* | 1 |
| *Pintomyia pacae* | 1 |
| *Psathyromyia runoides* | 1 |
| *Psathyromyia scaffi* | 1 |
| *Pintomyia serrana* | 1 |
| *Sciopemyia servulolimai* | 1 |
| *Micropygomyia trinidadensis* | 1 |
| Total | 3,218 |

**Table 2.** (Continued)

## Study area
### Suruwaha Indigenous Land (SIL)
The municipality of Tapauá is located on the banks of the Purus River. It has a population of 16,876 inhabitants and a total area of 84,946 km$^2$ [13]. The main economic activity is agriculture (cassava, jute and beans) and extractivism (nuts, rubber, wood, copaiba oil and andiroba). Livestock has become the main product in the last 10 years.

The Middle Purus region is in the state of Amazonas, in the southern part of the Amazon rainforest. It includes several conservation units, in addition to various indigenous lands, several of which are already officially recognized by the federal government, while others are currently in the process of formal legal demarcation.

According to the SIASI (Sistema de Informação da Atenção à Saúde Indígena), operated by the Secretaria Especial de Saúde Indígena (SESAI), the indigenous population of the Middle Purus is estimated at around 8,117 inhabitants. These are divided into 13 ethnic groups and distributed in 24 demarcated indigenous lands and another 6 non-demarcated (unidentified and/or delimited). The total territorial extension of the demarcated indigenous areas corresponds to 189,870.964 ha.

The Suruwaha Indigenous Land (SIL) is in the municipality of Tapauá. The SIL area comprises 239,070 ha and is located between the Riozinho River and the Coxodoá stream, both tributaries of the Cuniuá River, which in turn is a tributary of the Tapauá River that flows into the Purus River [19]. The SIL is surrounded to the west by the Deni Indigenous Land, and to the east by the Hi-Merimã Indigenous Land, of isolated indigenous peoples. The population consists of 171 people who were living in isolation and were contacted by missionaries in the 1980s [20]. The productive activities of the Suruwaha include agriculture, hunting, fishing, gathering and tool making [19]. Hunting is the most prestigious activity, and a good hunter not only kills many animals but must also have killed many tapirs, the most coveted game because of its size, which can feed many people [20].

ACL is endemic to the Amazon region, and the state of Amazonas contributed 20% (34,169 cases) of the total of cases (167,186) recorded for the north region of Brazil, where the Amazon is located, between 2002 and 2020 [5]. See Figure 1 for a map of the georeferenced occurrences in GBIF.

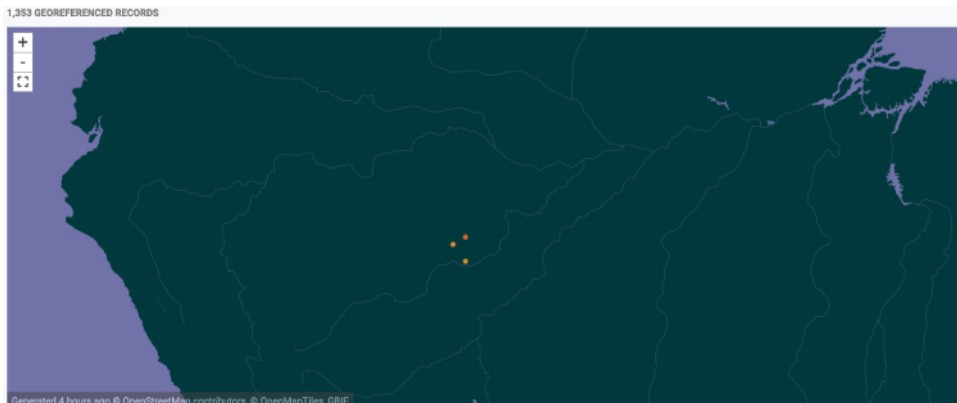

**Figure 1.** Interactive map of the georeferenced occurrences of sand flies from the Suruwaha Indigenous Land, hosted by GBIF [21]. https://www.gbif.org/dataset/d9f6d2bf-20d7-4b87-99ac-e902b34364e4

### *Wajãpi Indigenous Land (WIL)*

The Wajãpi Indigenous Land (WIL) extends between the basins of the Jari (to the west), Amapari (to the east) and Oiapoque (to the north) rivers. Official recognition by the Brazilian government occurred in 1996, with a territorial extension of 607,000 km$^2$. The area is dense tropical forest and in rugged relief, integrating the Tumucumaque Mountains complex. Currently, the Wajãpi number about 1,221 people, distributed in more than 80 villages [17].

The Wajãpi organize themselves into autonomous local groups called 'iwanã-ko', which are represented by a local group that has its origin in a specific region, where there are several villages. However, not all the people of an iwanã-ko live in the same region, because when marriages between people from different groups take place, one of the spouses starts living in the other's region [22], which may represent only a temporary physical settlement. The different groups occupy three spatial categories: the places of concentration: rural villages/dwellings; intermittent dispersal/settlements (mainly during the dry season – hunting, fishing and gathering camps); and 'koo kwerã' sites, which are wildlife reserves left undisturbed for wild animals to use and therefore, be hunted. This combination of social, political and ecological factors that determine the movements of concentration and dispersion of the Wajãpi in its territory seems to accompany the patterns of disease distribution in the TIW, so that the tendency to concentrate families close to the Perimetral Norte highway, which reaches TIW, seems to be related to the increased incidence of infectious diseases [18, 23].

The state of Amapá contributed 6% (10,125) of the total of cases (167,186) of ACL recorded for the north region of Brazil between 2002–2020 [5]. See Figure 2 for a map of the georeferenced occurrences in GBIF.

### Sand fly collection and processing

Sand flies were collected with unbaited US Centers for Disease Control (CDC)-like light traps operated between 5:00 pm and 6:00 am. Traps were placed in areas used by the indigenous people, such as swiddens, forest areas, hunting grounds, access trails and homes.

Insects were stored in microtubes containing 70% alcohol, and were sorted from the other insects collected either in the field or under a dissecting microscope in the laboratory.

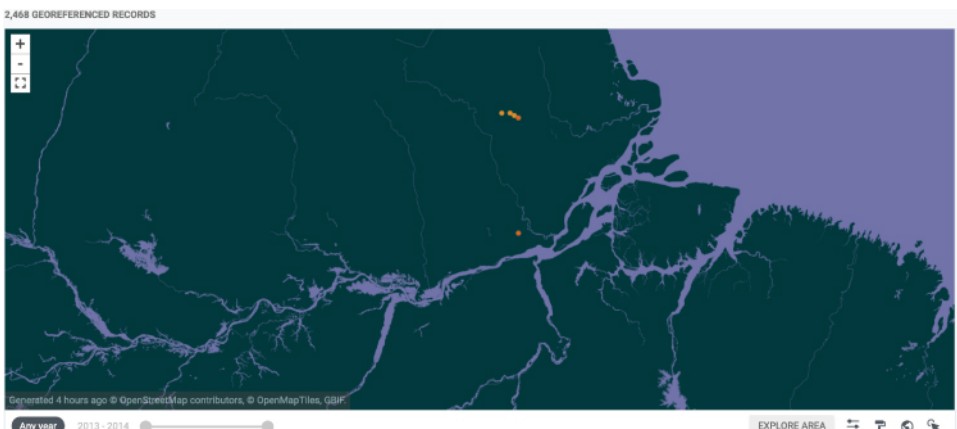

**Figure 2.** Interactive map of the georeferenced occurrences of sand flies from the Wajãpi Indigenous Land hosted by GBIF [24]. https://www.gbif.org/dataset/7140a42f-8ad6-43cd-b252-b5a791c8de4a

The insects were clarified and mounted on a slide to identify the species, using the morphological identification key by Galati [2]. A subsample of the insects will be deposited in the Coleção de Flebotomíneos of the Instituto René Rachou/FIOCRUZ-Minas (COLFLEB/FIOCRUZ).

## DATA VALIDATION AND QUALITY CONTROL

Insects were identified by keys available in the literature [2, 3] by experienced taxonomists. The dataset is in Darwin Core format; 41 terms are available for the Suruwaha dataset and 39 for the Wajãpí dataset. All mandatory fields are present and have undergone screening in the FIOCRUZ IPT. Metadata fields and datasets are also available on the online pages [5, 6].

## REUSE POTENTIAL

These data are important because they describe the distribution of sand flies collected at different sites in two indigenous lands in the Brazilian Amazon. The data can be used by different sectors of academia, government, civil society, and by non-governmental organizations. However, these data may be of particular importance to balance scientific knowledge with indigenous knowledge to improve health surveillance activities and adapt these to different ecosocial contexts with the participation of indigenous people, who better know their territories.

These data can be used to address challenges in leishmaniases control, and to better understand the epidemiology of this disease. Since control measures in Brazil are based on disease surveillance and monitoring in territorial units, which include biological and environmental characteristics, our dataset can contribute to a broader knowledge base. Our data provide occurrence records from locations that would not normally be surveyed by the public health system ACL control programs of the Brazilian government. These data can be used to model both vector and disease distribution in space and time, as well as provide clues on priority areas for ACL surveillance and control in these areas.

Our high-quality data provide an expert-validated list of sand fly species with up-to-date names from remote areas in the Brazilian Amazon.



## DATA AVAILABILITY

The data supporting this article are published through the FIOCRUZ – Oswaldo Cruz Foundation IPT and are available under a CC0 waiver from GBIF [21, 24].

## EDITOR'S NOTE

This paper is part of a series of Data Release articles working with GBIF and supported by the Special Programme for Research and Training in Tropical Diseases (TDR), hosted at the World Health Organization [25].

## DECLARATIONS
## LIST OF ABBREVIATIONS

ACL: American cutaneous leishmaniasis; CDC: Centers for Disease Control; GBIF: Global Biodiversity Information Facility; IPT: integrated publishing toolkit, SIL: Suruwaha Indigenous Land; WIL: Wajãpi Indigenous Land.

## ETHICAL APPROVAL

The collection licenses and permits for our studies were as follows: in the SIL, SISBIO collecting license, issue by the Sistema de Autorização e Informação em Biodiversidade (IBAMA) (39337-1) and FUNAI (08620.040969/2013-51); and in the WIL, SISBIO collecting license (37935-4), FUNAI (08620.030843/2014-59), Ethics Committee approval (CONEP - CAAE: 20188213.9.0000.5091), and IPHAN access to associated traditional knowledge for scientific research purposes (01450.008806/2014-14).

## CONSENT FOR PUBLICATION

Not applicable.

## COMPETING INTERESTS

The authors declare that they have no competing interests.

## FUNDING

This project was funded by Conselho Nacional de Desenvolvimento Científico e Tecnológico CNPq (grant numbers 474506/2012-6 and 404390/2012-9) – PHFS; Coordenação de Aperfeiçoamento de Pessoal de Nível Superior (Capes), and Fundação de Amparo à Pesquisa do Amapá – Fapeap (004/2013) – ESM.

## AUTHORS' CONTRIBUTIONS

PHFS: provision of resources, funding acquisition, conceptualization of the research, fieldwork, data collection and analysis, supervision, preparation of manuscript; DRCA: fieldwork, provision of resources, data collection; ESM: provision of resources, funding acquisition, data analysis, conceptualization of the research; JACB: sample preparation; MPF: fieldwork, conceptualization of the research, data collection; MEMS: fieldwork, provision of resources, data collection; MDGGA: sample preparation; SFM: sample preparation; TSC: sample preparation; VRA: sample preparation and identification; LAB: data curation, revision of manuscript.

## ACKNOWLEDGEMENTS

We thank the Suruwaha people for allowing us to stay with them and collect sand flies. To the Wajãpi Indigenous Council: Apina, Associação dos Povos Indígenas Wajãpi do Triangulo do Amapari - Apiwata, Associação Wajãpi Terra, Ambiente e Cultura - Awatac. To Instituto de Pesquisa e Formação Indígena (Iepé). To the Wajãpi for their support during fieldwork and contributions on sand fly collecting sites: Aikyry, Jatuta, Japu, Jawaruwa, Piriri, Jawapuku, Nazare, Marãte, Wajamã, Moratu, Wynamea, Wawa, Waiwai, Seremeté, Waraku, Majuware, Asurui, Wyrai, Janeanã, Roseno, Patiheu, Aka'upotyr, Savã, Joapiria, Jakyri, Parikura, Tukuruwe, Yrovaite, Kasiripinã (in memoriam), Paiki, Moroti, Tapenaiki, Tarakuasii, Pakitu, Patire, Kumaré, Keremeti, Tameri, Namirõ, Ripé, Sirara, Pasiku, Seki, Taresa, Karara, Ororiwõ, Jamano, Viseni, Inarina, Kenewe, Mika, Apamu, Patena, Kawãe, Paniu, Taruku, Siró, Kourupe, and all the Wajãpí Community. To Fundação Nacional do Índio, Secretaria Especial de Saúde Indígena – SESAI. To Alline Silva da Costa (Universidad Nacional de Rosario, Joana Cabral Oliveira (UNICAMP), Juliana Rosalen (Iepé), Luis Alberto Sabioni, Raimundo Nonato Picanço Souto (Unifap), Rodolfo G.A.V. Stumpp, Thais Jeniffer das Dores Cardoso, Volmir Zanni (LACEN/AP). And to Clara Baringo Fonseca (RNP - Rede Nacional de Ensino e Pesquisa).

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
